# Circular RNA circIGF2BP3 Promotes the Proliferation and Differentiation of Chicken Primary Myoblasts

**DOI:** 10.3390/ijms242115545

**Published:** 2023-10-24

**Authors:** Xiaotong Wang, Junyuan Lin, Zhenhai Jiao, Li Zhang, Dongxue Guo, Lilong An, Tingting Xie, Shudai Lin

**Affiliations:** College of Coastal Agricultural Sciences, Guangdong Ocean University, Zhanjiang 524088, China; wangxiaotong_0320@163.com (X.W.); jylin1016@gdou.edu.cn (J.L.); jiaozhenhai2022@163.com (Z.J.); zhangli761101@163.com (L.Z.); gdxovo@163.com (D.G.); anlilong@126.com (L.A.); ttxie@gdou.edu.cn (T.X.)

**Keywords:** *Yuexi frizzled feather chicken*, primary myoblasts, circIGF2BP3, proliferation, differentiation

## Abstract

The quality and quantity of animal meat are closely related to the development of skeletal muscle, which, in turn, is determined by myogenic cells, including myoblasts and skeletal muscle satellite cells (SMSCs). Circular RNA, an endogenous RNA derivative formed through specific reverse splicing in mRNA precursors, has the potential to influence muscle development by binding to miRNAs or regulating gene expression involved in muscular growth at the transcriptional level. Previous high-throughput sequencing of circRNA in chicken liver tissue revealed a circular transcript, circIGF2BP3, derived from the gene encoding insulin-like growth factor 2 mRNA binding protein 3 (IGF2BP3). In this study, we confirmed the presence of the natural circular molecule of circIGF2BP3 through an RNase R enzyme tolerance assay. RT-qPCR results showed high circIGF2BP3 expression in the pectoral and thigh muscles of *Yuexi frizzled feather chickens* at embryonic ages 14 and 18, as well as at 7 weeks post-hatch. Notably, its expression increased during embryonic development, followed by a rapid decrease after birth. As well as using RT-qPCR, Edu, CCK-8, immunofluorescence, and Western blot techniques, we demonstrated that overexpressing circIGF2BP3 could promote the proliferation and differentiation of chicken primary myoblasts through upregulating genes such as proliferating cell nuclear antigen (*PCNA*), cyclin D1 *(CCND1*), cyclin E1 (*CCNE1*), cyclin dependent kinase 2 (*CDK2*), myosin heavy chain (*MyHC*), myoblast-determining 1 (*MyoD1*), myogenin (*MyoG*), and *Myomaker*. In conclusion, circIGF2BP3 promotes the proliferation and differentiation of myoblasts in chickens. This study establishes a foundation for further investigation into the biological functions and mechanisms of circIGF2BP3 in myoblasts proliferation and differentiation.

## 1. Introduction

Circular RNA (circRNA) is an endogenous RNA generated through the specific reverse splicing of mRNA precursors [1]. Notably, circRNA exhibits remarkable resistance to RNase R enzymes when compared to linear mRNA transcripts [2]. CircRNAs are prevalent in cells and tissues, and regulates gene expression through various mechanisms at both the transcriptional and post-transcriptional levels [3]. Previous studies have found that circRNAs primarily function through the following mechanisms: (1) miRNA sponges. For example, ciRS-7 promotes the differentiation of goat skeletal muscle satellite cells (SMSCs) by binding to miR-7 [3], and circTitin (circTTN) enhances the proliferation and differentiation of bovine SMSCs by competitive binding to miR-432 [4]. (2) Transcriptional regulation. CircSepallata3 (circSEP3), derived from exon 6 of Sepallata3 (*SEP3*), can bind to specific DNA sites, forming an RNA-DNA complex that inhibits the transcription of *SEP3* [5]. (3) Coding potential. Many studies have reported that some circRNAs possess coding potential, resulting in protein translation [6,7,8]. (4) Protein binding. Interactions with proteins allow circRNAs to participate in various cellular activities. For instance, cerebellar degeneration-related protein 1 transcript (*CDR1as*) circRNA is tightly bound by Argonaute (AGO) [9]. 

Insulin-like growth factor 2 mRNA binding protein 3 (*IGF2BP3*) belongs to the insulin-like growth factor mRNA binding protein (*IGFBP*) family; its protein has the ability to bind to insulin-like growth factor 2 (*IGF2*) mRNA. It regulates critical aspects of *IGF2* mRNA, such as localization, stability, reverse transcription, and translation [10,11]. Previous research has revealed that downregulating *IGF2BP3* significantly delays differentiation and induces the proliferation of C2C12 myoblasts, underscoring its pivotal role in muscle development by promoting myoblast proliferation [12]. Furthermore, *IGF2BP3* is a candidate gene associated with intramuscular fat (IMF) deposition [13]. Research has observed that, in comparison to normal chickens, the mRNA expression of *IGF2BP3* gene in the insulin pathway of skeletal muscle in dwarf chickens is decreased, leading to a significant increase in IMF deposition [13]. Additionally, miR-9-5p was found to regulate the proliferation and differentiation of chicken SMSCs by targeting both *IGF2BP3* and *IGF2* [14]. The chicken *IGF2BP3* gene is located on chromosome 2 and spans a total length of 10.74 kilobases (kb), consisting of 18 exons. In this study, we analyzed the circRNA high-throughput sequencing data previously conducted on chicken liver tissue by our team, and discovered circular transcripts originating from the *IGF2BP3* gene (named circIGF2BP3). However, it remains unclear whether circIGF2BP3 plays a role in regulating chicken skeletal muscle development. Therefore, we conducted an investigation encompassing sequence conservation, circular structure characteristics, expression patterns, and the potential impact on the proliferation and differentiation of primary chicken myoblasts.

## 2. Results

### 2.1. Circular Features of the Chicken CircIGF2BP3

Our previous high-throughput sequencing has shown that the chicken *IGF2BP3* gene could generate a circular transcript. Its presence in chickens was verified by PCR using specific primers targeting the splice junction. As expected, a target fragment was observed at the 236 bp position, which corresponds to the length of the circIGF2BP3 adapter fragment (Figure 1A). This provides initial evidence of a reverse splicing product originating from the *IGF2BP3* gene. Furthermore, we initially reported that circIGF2BP3 is a circRNA composed of exons 6 and 10 of the *IGF2BP3* gene (Figure 1B), with a total length of 811 nucleotides (nt), and contains four open reading frames (ORFs) (Figure 1C,D).

To confirm the circular structure of circIGF2BP3, RNA extracted from the liver tissue of *Yuexi frizzled feather chickens* was reverse transcribed into complementary DNA (cDNA) using both untreated and RNase R-treated samples, respectively. The results of a real-time fluorescence quantitative PCR (RT-qPCR) indicated that the relative expression of *IGF2BP3* mRNA significantly decreased in the RNase R treatment group, while the relative expression of circIGF2BP3 remained unchanged (Figure 1E). In addition, our RT-qPCR analysis revealed that the expression of *IGF2BP3* mRNA was unchanged in both the oligo-d(T)20VN and random primer groups. However, the expression of circIGF2BP3 was significantly higher in the random primer reversed group compared to the oligo-d(T)20VN group (Figure 1F). It was further confirmed that circIGF2BP3 lacks a Poly (A) tail at its 3′ end, verifying its circular molecule characteristic. Results from nuclear-cytoplasmic separation and RT-qPCR indicated that circIGF2BP3 exhibited a relatively balanced distribution between the nucleus and cytoplasm of chicken myoblasts. In contrast, the expression of its linear transcript, *IGF2BP3*, was significantly higher in the cytoplasm than that in the nucleus (Figure 1G).

### 2.2. Spatiotemporal Expression Pattern of Chicken CircIGF2BP3

#### 2.2.1. Tissue Expression Profile of CircIGF2BP3

RNA was extracted from the heart, liver, spleen, lung, kidney, and pectoral and thigh muscles of *Yuexi frizzled feather chickens* at three developmental stages: embryonic ages 14 and 18 (E14, E18), and 7 weeks old (7W). This was then reverse transcribed into cDNA. Subsequently, the expression patterns of both circIGF2BP3 and *IGF2BP3* mRNA in chickens were investigated using RT-qPCR. The results revealed a similar tissue-specific expression pattern for both circIGF2BP3 and *IGF2BP3* mRNA. High expression levels of both were observed in the pectoral and thigh muscle tissues of *Yuexi frizzled feather chickens* at E14, E18, and 7W, while low expression levels were found in the lung and spleen tissues (Figure 2A–F).

#### 2.2.2. Temporal Expression Pattern of CircIGF2BP3

RNA was extracted from the pectoral and thigh muscle tissues of *Yuexi frizzled feather chickens* at various time points before and after birth. RT-qPCR results showed an increasing expression pattern of circIGF2BP3 in the pectoral and thigh muscles of *Yuexi frizzled feather chickens* before birth, followed by a decrease after hatching (Figure 3A,C). The highest expression level of *IGF2BP3* mRNA was observed in tissue from the pectoral and thigh muscles of Yuexi frizzed feather chickens at E18 (Figure 3B,D).

### 2.3. CircIGF2BP3 Promotes the Proliferation and Differentiation of Chicken Primary Myoblasts

#### 2.3.1. CircIGF2BP3 Expression in the Proliferation and Differentiation of Myoblasts

The RNA was isolated from myoblasts at different stages, including 50% and 100% cell confluence, as well as during 1–5 days of differentiation. RT-qPCR results demonstrated significantly higher expression of circIGF2BP3, *IGF2BP3*, and *IGF2* mRNAs during days 1–5 of differentiation (DM1–5), pre-DM4, and at DM1 compared to the conditions at 50% cell confluence (Figure 4A–C). Additionally, the expression levels of the differentiation marker gene, myosin heavy chain (*MyHC*), increased during myoblast differentiation (Figure 4D).

#### 2.3.2. CircIGF2BP3 Promotes the Proliferation of Myoblasts

To analyze the effect of circIGF2BP3 on the proliferation of chicken primary myoblasts, we initially conducted a CCK-8 assay. The results showed that primary myoblasts overexpressing circIGF2BP3 exhibited significantly higher proliferation compared to control cells (Figure 5A). In addition, the EdU assay revealed a significant increase in the number of EdU-positive cells after circIGF2BP3 overexpression compared to control cells (Figure 5B). The findings of the flow cytometry assay revealed a significant increase in the proportion of cells in the S-phase and a decrease in the G0/G1 phase after circIGF2BP3 overexpression (Figure 5C). Furthermore, RT-qPCR and Western blot analyses revealed a significant increase in both the relative mRNA and protein expression levels of proliferation marker genes, such as cyclin D1 (*CCND1*), proliferating cell nuclear antigen (*PCNA*), and cyclin dependent kinase 2 (*CDK2*), after transfection with pCD2.1-circIGF2BP3 for 48 h compared to the control group (Figure 5D,E). Taken together, these findings collectively indicate that circIGF2BP3 can promote the proliferation of chicken primary myoblasts.

#### 2.3.3. CircIGF2BP3 Promotes Myoblast Differentiation

For investigating the influence of circIGF2BP3 on primary myoblast differentiation, we conducted immunofluorescence, RT-qPCR, and Western blot analyses. The immunofluorescence analysis targeting MyHC revealed a significant increase in the relative myotube area after circIGF2BP3 overexpression (Figure 6A). Furthermore, the RT-qPCR analysis indicated a remarkable increase in the relative mRNA expression levels of four muscle differentiation marker genes, including myoblast-determining 1 (*MyoD1*), myogenin (*MyoG*), *MyHC*, and *Myomaker*, compared to the control group (Figure 6B). Western blot results demonstrated a significant upregulation in the protein levels of MyoD1 and MyHC after overexpressing circIGF2BP3 (Figure 6C). These findings illustrate that circIGF2BP3 plays a role in promoting the differentiation of chicken primary myoblasts.

## 3. Discussion

In recent years, circular RNA has received much attention due to its unique circular structure. Increasing evidence points towards the pivotal role of circRNAs in various biological processes, including skeletal muscle development [15]. Furthermore, IGF2BP3, a secretory protein capable of binding to *IGF2* mRNA, has been shown to have close associations with animal muscle development and myoblast proliferation [12,13,14]. In our initial research, we conducted sequencing on liver tissue from *Yuexi frizzled feather chickens* and identified multiple differentially expressed circRNAs, including circIGF2BP3, which is derived from the *IGF2BP3* gene. However, the role of circIGF2BP3 in skeletal muscle development in both poultry and mammals has remained unexplored, and its underlying molecular mechanism remains unclear. Therefore, we decided to focus on chicken circIGF2BP3, which is a particularly intriguing molecule. We determined that it originates from the reverse splicing of exons 6 to 10 of the *IGF2BP3* mRNA. Notably, it has been observed that the human *IGF2BP3* gene can also generate another circIGF2BP3. This circIGF2BP3 exerts an immunosuppressive effect on non-small cell lung cancer cells (NSCLC) by increasing the expression of plakophilin 3 (*PKP3*) through competitive binding to miR-328-3p and miR-3173-5p [16].

CircRNA is well-known for its covalently closed loop structure, which lacks both a 5′ cap and a 3′ poly (A) tail. Additionally, it demonstrates remarkable resistance to the RNase R enzyme [17]. Through divergent reverse-transcription PCR and RNase R treatment, we confirmed that circIGF2BP3 is a stable exonic circRNA generated by the *IGF2BP3* gene, with a length of 811 nt. Simultaneously, our investigation revealed that chicken circIGF2BP3 is almost equally distributed within both the nucleus and cytoplasm of myoblasts, whereas the *IGF2BP3* mRNA is predominantly distributed in the cytoplasm. Furthermore, it has been documented that the *IGF2BP3* gene can transcribe a variety of circIGF2BP3 molecules in various species, including *humans* [16], *mice* [18], *pigs* [19], and *ducks* [20]. We also identified the presence of circIGF2BP3 in rats by scrutinizing circRNA databases. In avian animals, both duck [20] and chicken (our group) embryonic muscle tissues exhibited the expression of 4 and 6 distinct forms of circIGF2BP3, respectively. To broaden our perspective, we conducted homology analyses on circIGF2BP3 sequences from other species that align with the chicken circIGF2BP3 (exons 6–10) (Appendix A). Unexpectedly, the result underscored a substantial conservation of homology, surpassing 75% across diverse species, as shown in the Appendix A. This high degree of conservation emphasizes the significance of circIGF2BP3’s role and suggests a potential correlation in its biological functions among different species.

Muscle fibers are the basic composition of skeletal muscle. The total amount of muscle in animals is primarily influenced by various factors, such as the total number of muscle fibers (hyperplasia), their thickness (hypertrophy) and their types [21]. In poultry, the number of muscle fibers remains nearly constant from birth, and post-birth muscle development depends on the proliferation and differentiation of embryonic myoblasts and the function of early SMSCs, which could result in myofiber hypertrophy and transitions in myofiber type, leading to the enlargement and thickening of skeletal muscle [22]. Importantly, the 14th embryonic age is a critical period for the proliferation and differentiation of skeletal muscle myoblasts, ultimately leading to the fusion of myotubes and the formation of muscle fibers, a process that is essentially completed around the 20th embryonic age [23]. In the current study, we successfully confirmed elevated expression levels of both circIGF2BP3 and its parent gene’s (*IGF2BP3*) linear mRNA in the pectoral and thigh muscles of *Yuexi frizzled feather chickens* at E14, E18 and 7W, compared to other tissues. Meanwhile, our observations indicated that the relative expression levels of both circIGF2BP3 and *IGF2BP3* mRNA in pectoral and thigh muscle tissues of *Yuexi frizzled feather chickens* reached their peak during the early embryonic stage (E10–E18), and subsequently declined after birth. The results strongly suggest that circIGF2BP3 may play a crucial role in the intricate process of chicken’s skeletal muscle development. 

Therefore, in order to gain a deeper understanding of the precise function of circIGF2BP3 in chicken’s skeletal muscle development, we conducted a series of experiments. According to the findings of multiple analyses, including RT-qPCR, Edu, CCK-8, immunofluorescence and Western blot assays, it was found that circIGF2BP3 is essential in promoting the proliferation and differentiation of chicken primary myoblasts. Previous studies have revealed the regulatory roles of several circRNAs in chicken myogenesis. For instance, circSupervillin (circSVIL), was found to act as a miR-203 sponge, promoting myoblasts differentiation and proliferation [24]. Furthermore, a novel circular RNA, generated by the fibroblast growth factor receptor 2 gene (circFGFR2), has been shown to promote the proliferation and differentiation of chicken primary myoblasts by acting as a sponge for miR-133a-5p and miR-29b-1-5p [25]. Additionally, circIntersectin 2 (circITSN2) has been found to enhance the proliferation and differentiation of chicken embryonic myoblasts by binding to miR-218-5p [26]. Moreover, the circular protein phosphatase 1 regulatory subunit 13B (circPPP1R13B) has been reported to promote the proliferation and differentiation of chicken SMSCs by binding to miR-9-5p and activating the IGF2-PI3K/Akt signaling pathway [27]. Similarly, a novel protein encoded by the circular family with a sequence similarity to 188 member B (circFAM188B), known as circFAM188B-103aa, has been shown to promote the proliferation, but inhibit the differentiation of chicken SMSCs [28]. In our previous analysis, we used the RNAhybrid online tool to predict the potential target sites of circIGF2BP3, and found multiple miRNA target sites, including miR-15a, miR-15b-5p, and miR-200b-3p. Previous research has found that miR-15a directly interacts with the 3′UTR of Cyclin D1 (*CCND1*) mRNA, playing a crucial role in the post-transcriptional regulation of *CCND1* expression [29]. However, whether circIGF2BP3 functions as a sponge for these miRNAs to regulate chicken myoblast proliferation and differentiation still requires further research. Intriguingly, in the current study, bioinformatic analysis showed that circIGF2BP3 contains multiple ORFs, and Western blot results confirmed that the longest ORF is capable of encoding a protein. In conclusion, the present study revealed that circIGF2BP3 promotes the proliferation and differentiation of chicken primary myoblasts. We eagerly anticipate further research to elucidate whether circIGF2BP3 functions as a miRNA sponge or as a translated protein, thus regulating myoblasts proliferation and differentiation.

## 4. Materials and Methods

### 4.1. Ethics Standards

The hatching of fertilized eggs and the breeding management of *Yuexi frizzled feather chickens* were conducted at the experimental station of Guangdong Ocean University. The Institutional Review Board of Guangdong Ocean University in Zhanjiang, China (SYXK-2021-0154), authorized the use of animals and procedures for this study.

### 4.2. Animal

*Yuexi frizzled feather chickens* used in this study were purchased from Nanxia Village Chicken Farm in Mazhang District, Zhanjiang City, Guangdong Province. The fertilized eggs of Yuexi frizzled feather chicken, which were used for cell separation, were artificially inseminated and incubated by our team.

### 4.3. Full-Length Cloning and Validation of Loop Formation in CircIGF2BP3

Two primer pairs were designed, with each pair targeting one side of the circIGF2BP3 junction position. The full-length amplification primer (circIGF2BP3-full) was used to analyze the complete sequence of circIGF2BP3, while the circular (junction) position primer (circIGF2BP3-C) was employed to detect the presence of circIGF2BP3. All primers were designed using Primer-BLAST (https://www.ncbi.nlm.nih.gov/tools/primer-blast/ (accessed on 13 June 2022)). Subsequently, the products resulting from PCR amplification with these primers were subjected to Sanger sequencing. Detailed primer information is provided in Table 1.

To assess the stability of circIGF2BP3, the total RNA was subjected to digestion at 37 °C for 15 min, followed by incubation at 70 °C for 10 min. Subsequently, the total RNA was reverse transcribed into cDNA using either oligo-d(T)20VN or random primers (Vazyme, Nanjing, China), respectively. The expression of circIGF2BP3 was quantified through RT-qPCR, with *β-actin* serving as the internal reference gene (Table 1).

### 4.4. Construction of Chicken CircIGF2BP3 Overexpression Vector

The full-length sequence of circIGF2BP3, containing a 3× Flag tag protein, was cloned into the pCD2.1-ciR overexpression vector (named pCD2.1-circIGF2BP3), which was constructed by WuHan Genecreate Biological Engineering Co., Ltd. The control vector used in this study was the empty pCD2.1-ciR, which was purchased from Gsay Biotech.

### 4.5. Cell Culture and Transfection

Primary myoblasts from chickens were collected following the procedures outlined in the previously reported study by Duan et al. [30]. Leg muscle tissues from E10.5 chickens were isolated, chopped with scissors, and washed with Phosphate Buffered Saline (PBS, Gibco, Waltham, MA, USA). They were digested with 0.25% trypsin (Gibco) in a 37 °C cell culture incubator for 20 min. Then, cells were collected using centrifugation at 1500 rpm, and the supernatant was discarded. The cells were resuspended in RPMI Medium 1640 (RPMI 1640, Gibco), supplemented with 20% fetal bovine serum (FBS, Gibco), and 1% penicillin-streptomycin (Gibco). They were cultured at 37 °C for 40 min to eliminate other cell types, such as fibroblasts. This process was repeated twice. Subsequently, a medium containing 20% fetal bovine serum (FBS) was used in regular culture conditions.

The F2 primary myoblasts were seeded into 6-well plates and cultured in RPMI 1640 medium with 20% FBS until they reached 85% confluence. Subsequently, the cells were transfected with either the empty vector pCD2.1-ciR or the overexpression vector pCD2.1-circIGF2BP3 using Lipofectamine^®^ 3000 (Thermo Fisher, Waltham, MA, USA) according to the manufacturer’s protocol. After 48 h of transfection, the cells were collected for a cell proliferation test. To induce cell differentiation, the RPIM 1640 medium containing 20% FBS was replaced with a 2% horse serum medium after 48 h of transfection. The cells were then cultured in a sterile incubator at 37 °C with 5% CO_2_.

### 4.6. RNA Extraction and Quantitative Real-Time PCR (RT-PCR)

Total RNA from both tissues and cells was extracted following the guidelines of the HiPure Universal RNA Mini Kit (Men, Guangzhou, China). The total RNA was then reverse transcribed into cDNA using the HiScript^®^ RT SuperMix for qPCR kit (Vazyme, Nanjing, China), following the provided instructions. The reaction program consisted of incubation at 42 °C for 2 min, 37 °C for 15 min, and 85 °C for 5 s.

Quantitative RT-PCR (Vazyme) was performed following the manufacturer’s protocols, with primer details provided in Table 2. The reaction mixture consisted of 2 μL of cDNA template, 0.4 μL each of forward and reverse primers, 10 μL of 2 × SYBR Green qPCR Mix, and 7.2 μL of double-distilled water. The reaction conditions consisted of an initial denaturation at 95 °C for 30 s, followed by 40 cycles of 95 °C for 5 s and 60 °C for 30 s. For the melting curve analysis, the procedure involved 95 °C for 15 s, 60 °C for 1 min, and 95 °C for 1 s. The mRNA abundance of each gene was determined using the CFX 96-TouchTM real-time PCR detection system (Bio-Rad, Harkles, CA, USA). The relative expression levels of the relevant genes were normalized using β-actin (Table 1) as the internal reference gene. The details of the quantitative primer are shown in Table 2.

### 4.7. Nuclear-Cytoplasm Separation

Nucleocytoplasmic separation was performed using the cytoplasmic and nuclear RNA extraction kit (AmyJet Scientific, Wuhan, China). The cytoplasmic and nuclear RNA was extracted and then reverse transcribed into cDNA following the method outlined in Section 4.6. The relative expression levels of circIGF2BP3 and IGF2BP3 mRNA were quantified by RT-qPCR using primers specific to circIGF2BP3 and IGF2BP3-201. β-actin and Sno-U6 genes were used as reference genes for the cytoplasmic and nuclear RNA, respectively. The primer information is listed in Table 2.

### 4.8. CCK-8 Assay

Cells were seeded in 96-well plates and transfected with both the empty vector pCD2.1-ciR and the overexpression vector pCD2.1-circIGF2BP3. Cell proliferation was assessed using the Cell Count Kit-8 (CCK-8) from GlpBio (Montclair, CA, USA). After incubation at 37 °C for 1 h, the absorbance at 450 nm was measured. This experiment was repeated eight times, with measurements taken at 0, 24, 48, 72, and 96 h post-transfection.

### 4.9. EdU Assay

After 48 h of transfection with both the empty vector pCD2.1-ciR and the overexpression vector pCD2.1-circIGF2BP3, cells were incubated, fixed, permeabilized, and stained following the instructions of the C10310 EdU Apollo In Vitro Imaging Kit (RiboBio, Guangzhou, China). Four random regions of each well were then photographed using fluorescence microscopy to assess the number of stained cells.

### 4.10. Flow Cytometry

Cells were seeded in 6-well plates and transfected with both the empty vector pCD2.1-ciR and the overexpression vector pCD2.1-circIGF2BP3. After 48 h, cells were harvested and stained with 0.5 mL of PI/RNase Staining Buffer (Coolaber, Beijing, China) per cell sample tube. Flow cytometry was used to analyze and quantify the stained cells.

### 4.11. Immunofluorescence

Primary myoblasts from the F2 generation were seeded into a 12-well plate and transfected when they reached approximately 90% confluence. After 48 h of transfection, when the cell confluence exceeded 95%, differentiation was induced by adding 2% pregnant horse serum (PMSG, Gibco) to the RPMI 1640 medium. After 3 days of differentiation induction, cells were fixed with 4% pre-cooled paraformaldehyde (Beyotime, Shanghai, China) at room temperature for 20 min. After fixation, the cells were washed with PBS for 5 min. To facilitate permeabilization, the cells were treated with 0.1% Triton X-100 for 10 min at room temperature. Then, an immunofluorescence blocking solution (Sangon Biotech, Shanghai, China) was applied for 1 h at room temperature. The primary rabbit antibody against MyHC (Proteintech, Wuhan, China) was subsequently incubated overnight at 4℃. Following incubation with the primary antibody, the cells were treated with FITC-labeled goat anti-rabbit IgG (Beyotime) and incubated for 2 h at room temperature in the absence of light. Subsequently, DAPI (4’,6-diamidino-2-phenylindole; Beyotime) was used for nuclear staining. Finally, the stained cells were photographed using a fluorescence microscope (Olympus, 6E07981, Tokyo, Japan).

### 4.12. Western Blot

After 48 h of transfection with both the empty vector pCD2.1-ciR and the overexpression vector pCD2.1-circIGF2BP3, protein was extracted from the myoblasts.

Antibodies used for the Western blot analysis included anti-IGF2, anti-PCNA, anti-CDK2, anti-CCND1, anti-MyoD1 (Abmart, Shanghai, China, diluted 1:500), and anti-MyHC (Proteintech, diluted 1:500).

### 4.13. Statistical Analysis

Statistical analyses were performed using SPSS 19.0 Statistical software (SPSS, Inc., Chicago, IL, USA). Each experiment was conducted at least three times for reliability. Data were presented as means ± standard error of the mean (SEM). The Student’s *t*-test was used for two-group comparisons, with a significance threshold of *p* < 0.05 applied to determine statistical significance. 

## Figures and Tables

**Figure 1 ijms-24-15545-f001:**
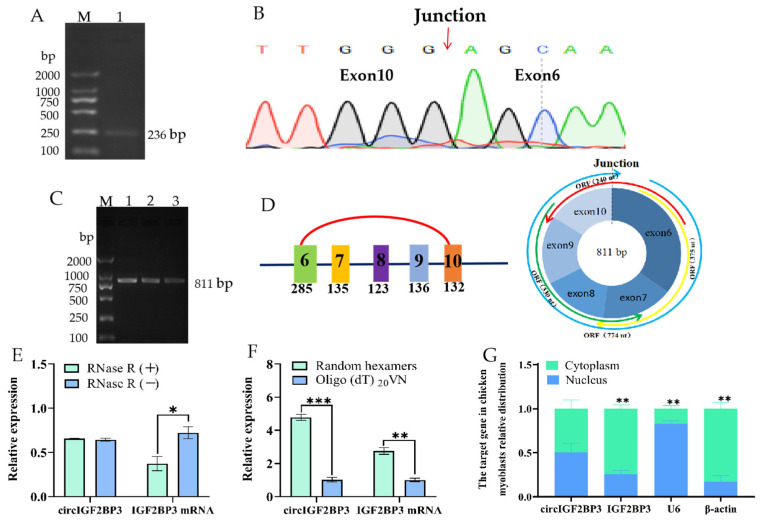
Circular features of the chicken circIGF2BP3. (**A**,**B**) Electrophoresis and sequencing results of the PCR product at the circIGF2BP3 junction site, respectively; (**C**,**D**) electrophoresis results and structural diagram of the full-length circIGF2BP3, respectively; (**E**) impact of RNase R treatment on circIGF2BP3 abundance and insulin-like growth factor 2 mRNA binding protein 3 (*IGF2BP3*) mRNA expression; (**F**) effect of different reverse transcription primers on the expression of circIGF2BP3 and *IGF2BP3* mRNA; (**G**) distribution of circIGF2BP3 and *IGF2BP3* mRNA in myoblasts. Data is presented as mean ± SEM (*n* = 4). * *p* < 0.05; ** *p* < 0.01; *** *p* < 0.001.

**Figure 2 ijms-24-15545-f002:**
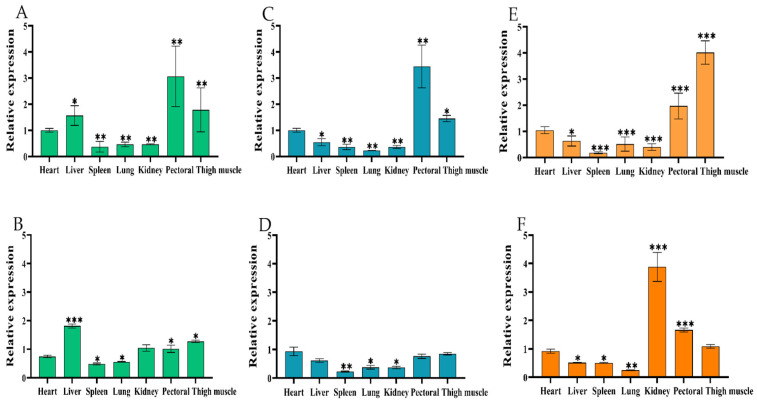
Tissue expression profiles of chicken circIGF2BP3 and *IGF2BP3* mRNA. (**A**,**C**,**E**) Expression patterns of circIGF2BP3 in *Yuexi frizzled feather chickens* at E14, E18, and 7W, respectively; (**B**,**D**,**F**) expression patterns of *IGF2BP3* mRNA in *Yuexi frizzled feather chickens* at E14, E18, and 7W, respectively. Data is presented as mean ± SEM (*n* = 4). * *p* < 0.05; ** *p* < 0.01; *** *p* < 0.001.

**Figure 3 ijms-24-15545-f003:**
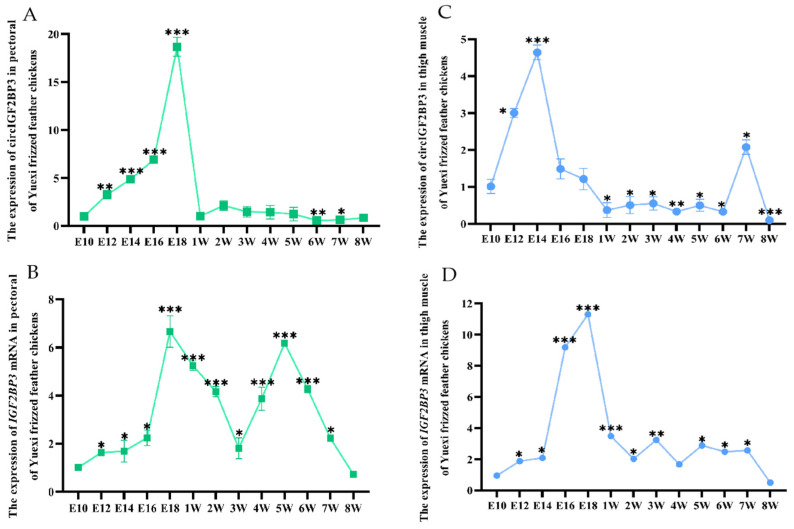
The temporal expression profile of circIGF2BP3 and *IGF2BP3* mRNA in chicken muscle tissue. (**A**,**C**) The expression pattern of circIGF2BP3 in the pectoral and thigh muscles of chickens at E10, E12, E14, E16, E18, and 1–8W, respectively; (**B**,**D**) the expression pattern of *IGF2BP3* mRNA in the pectoral and thigh muscles of chickens at E10, E12, E14, E16, E18, and 1-8W, respectively. Data is presented as mean ± SEM (*n* = 4). * *p* < 0.05; ** *p* < 0.01; *** *p* < 0.001.

**Figure 4 ijms-24-15545-f004:**
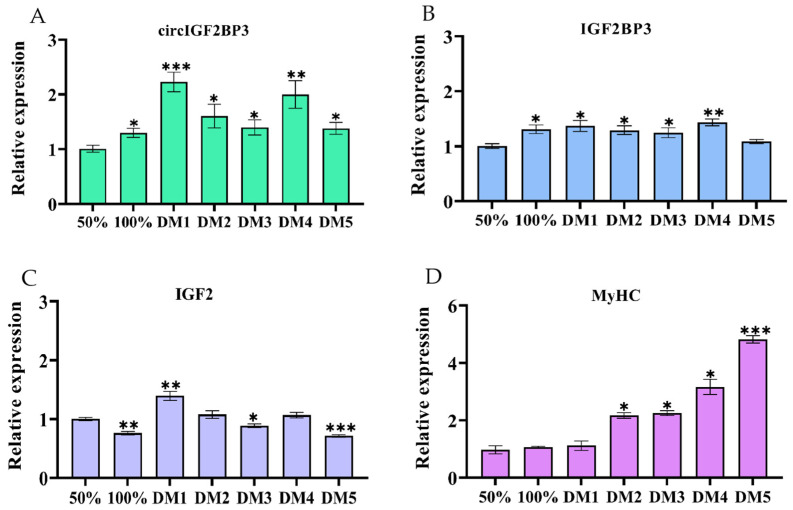
The expression levels of circIGF2BP3, *IGF2BP3*, insulin-like growth factor 2 (*IGF2*), and myosin heavy chain (*MyHC*) mRNAs during the proliferation and differentiation of primary myoblasts in chickens. (**A**–**D**) The relative expression levels of circIGF2BP3, *IGF2BP3*, *IGF2*, and *MyHC* mRNAs were analyzed during the proliferation and differentiation of chicken primary myoblasts, respectively. Data is presented as mean ± SEM (*n* = 4). * *p* < 0.05; ** *p* < 0.01; *** *p* < 0.001.

**Figure 5 ijms-24-15545-f005:**
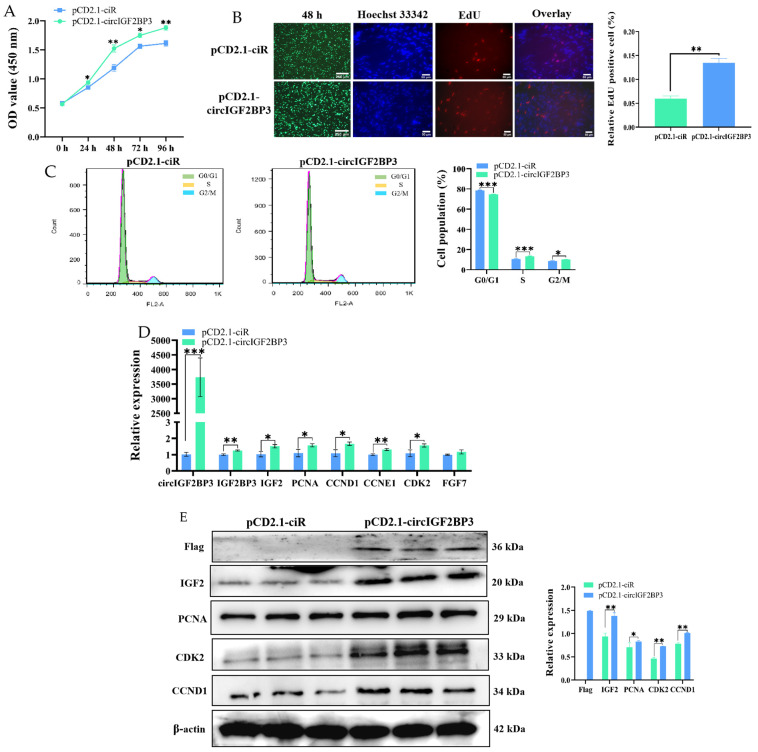
Effect of circIGF2BP3 on the proliferation of primary myoblasts (Scale bar: 250 μm and 50 μm). (**A**) Results of the CCK-8 assay on chicken primary myoblasts after transfection with pCD2.1-circIGF2BP3 and pCD2.1-ciR (NC); (**B**) results of the EdU assay on primary myoblasts 48 h after transfection with pCD2.1-circIGF2BP3 and pCD2.1-ciR (NC) (40×, 200× magnification); (**C**) cell cycle analysis of primary myoblasts 48 h after circIGF2BP3 overexpression and transfection with pCD2.1-ciR; (**D**) relative expression levels of circIGF2BP3, *IGF2BP3*, *IGF2*, proliferating cell nuclear antigen (*PCNA*), cyclin D1 (*CCND1*), cyclin E1 (*CCNE1*), cyclin dependent kinase 2 (*CDK2*), and fibroblast growth factor 7 (*FGF7*) mRNAs in primary myoblasts 48 h post-circIGF2BP3 overexpression and transfection with pCD2.1-ciR; (**E**) protein levels of circIGF2BP3 and different proliferation marker genes after the overexpression of circIGF2BP3. Data is presented as mean ± SEM (*n* = 3, 4, or 6). * *p* < 0.05; ** *p* < 0.01; *** *p* < 0.001.

**Figure 6 ijms-24-15545-f006:**
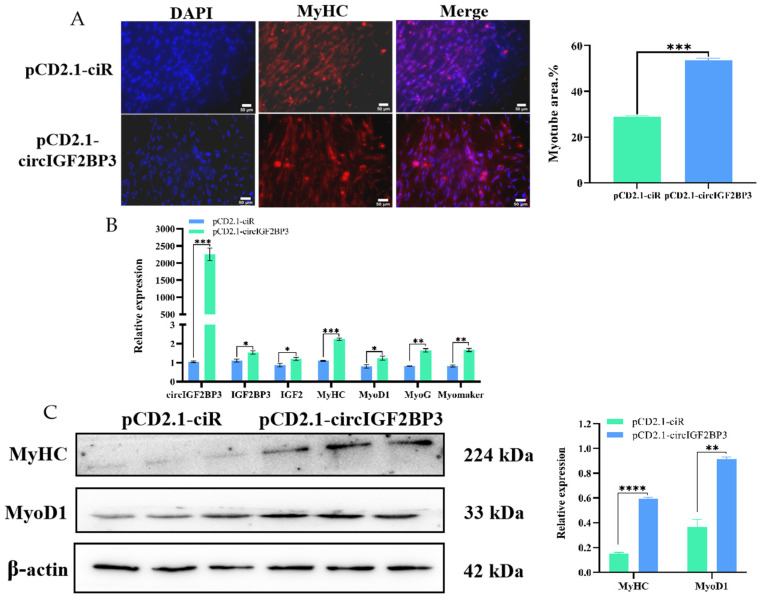
Effect of circIGF2BP3 on the differentiation of primary myoblasts (Scale bar: 50 μm). (**A**) MyHC immunofluorescent staining of primary myoblasts at 200× magnification. MyHC: indicated in red, serves as a molecular marker for myogenesis; DAPI staining in blue highlights cell nuclei; Merge illustrates the fusion of primary myoblasts into myotubes; the relative myotube area (%) after transfection with pCD2.1-circIGF2BP3 and pCD2.1-ciR (NC); (**B**) relative expression levels of circIGF2BP3, *IGF2BP3*, *IGF2*, *MyHC*, myoblast-determining 1 (*MyoD1*), myogenin (*MyoG*), and *Myomaker* mRNAs in primary myoblasts after 3 d of circIGF2BP3 overexpression and induced differentiation; (**C**) protein levels of myogenic marker genes after circIGF2BP3 overexpression and transfection with pCD2.1-ciR. Data is presented as mean ± SEM (*n* = 3 or 4). * *p* < 0.05; ** *p* < 0.01; *** *p* < 0.001; **** *p* < 0.0001.

**Table 1 ijms-24-15545-t001:** Primers used for full-length cloning and loop-forming validation.

Primer Name	Forward Primer (5′-3′)	Reverse Primer (5′-3′)	Annealing Temperature (°C)	Product Length (bp)
circIGF2BP3-C	CTGAATGCCTTGGGTCTGTTCC	GGCCCCTCTGTCCAAATCCA	64.5	811
circIGF2BP3-full	CCCAAATGGTGGATATGAAG	AGCAATAGAAAAACTGAACG	58	236
β-actin	ACTGACCGCGTTACTCCC	GCAACCATCACACCCTGATGTC	60	165

**Table 2 ijms-24-15545-t002:** Primers used for quantitative real-time PCR.

Primer Name	Forward Primer (5′-3′)	Reverse Primer (5′-3′)	Annealing Temperature (°C)	Product Length (bp)
circIGF2BP3	TCTGAATGCCTTGGGTCTGTT	GTCCAAATCCACGTCGTCCC	64.5	228
IGF2BP3-201	CTGCTGCTGCTTCATATCCACCATTT	CTCAGCTTGGCATCTGGTCCTTC	60	184
IGF2-202	CTGGCCTATGCGTTGGATTCAG	GTTGATCCTCCTGTTATTTCGTCCC	60	147
PCNA	CTCTGAGGGCTTCGACACCT	ATCCGCATTGTCTTCTGCTCT	60	133
CCND1	AACCCACCTTCCATGATCGC	CTGTTCTTGGCAGGCTCGTA	60	168
CCNE1	TTACGCTGTCCCCTGTTGAC	CCCAATTCCCACGCTACACT	60	265
CDK2	GTACAAGGCCCGGAACAAGG	TTCTCCGTGTGGATCACGTC	60	159
FGF7	AAAGACAGAAGGCAGGTCGG	TGCAAGCTAAAGATATAGTGCCCA	60	193
MyoD1	GCTACTACACGGAATCACCAAAT	CTGGGCTCCACTGTCACTCA	58	200
MyoG	CGGAGGCTGAAGAAGGTGAA	CGGTCCTCTGCCTGGTCAT	60	320
MyHC	CTCCTCACGCTTTGGTAA	TGATAGTCGTATGGGTTGGT	54	213
Myomaker	TGGGTGTCCCTGATGGC	CCCGATGGGTCCTGAGTAG	60	135
Sno-U6	CTCGCTTCGGCAGCACA	AACGCTTCACGAATTTGCGT	60	222

## Data Availability

No new data were created in this study.

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
