# Peer review of "Circular RNA circIGF2BP3 Promotes the Proliferation and Differentiation of Chicken Primary Myoblasts"

_ijms, 2023, doi:10.3390/ijms242115545_

Round 1
Reviewer 1 Report
The authors have found the presence of a circular transcript named circIGF2BP3 derived from the insulin-17 like growth factor 2 mRNA binding protein 3 (IGF2BP3) gene from their previous circRNA study on chicken liver. They show that circIGF2BP3 is formed by lining of Exon 6 to Exon 10. The authors claim that circIGF2BP3 promote proliferation and differentiation. Proliferation and differentiation are two alternative pathways for myoblast. Expression of genes which promote myoblast proliferation are usually turned down during differentiation and expression of genes which promote differentiation are turned on during differentiation. Turning transcription on and off transcription is possible for endogenous promoter driven expression but not possible for genes expressed from exogeneous promoter (I could not find the promoter which is used to overexpress circIGF2BP3). It is possible that circIGF2BP3 may promote proliferation by acting as eg Sponge for miRNA and promote differentiation using the Coding Potential of the circIGF2BP3 or vice versa. Thus, it is important that the authors first establish the coding potential of circIGF2BP3 by western blot analysis.
Comments:
1) Sequence Conservation of the Chicken CircIGF2BP3 is not necessary or should be moved to supplementary results.
2) Tissue Expression Profile of CircIGF2BP3 (Fig 3) and Temporal Expression Pattern of CircIGF2BP3 (Fig 4) three use a non-traditional method to show significance. Please use the traditional method of *, ** and *** to show the significance. Both only look at transcript level and do not provide any insight into the function of circIGF2BP3. Thus, these figures should be moved to supplementary or removed completely, unless the authors analyse the significance of the differences in the expression profile of CircIGF2BP3 and IGF2BP3.
3) The authors when on to show that CircIGF2BP3 Promotes the Proliferation and Differentiation of Chicken Primary Myoblasts but the figures are missing important controls.
4) In Figure 5 the authors show the expression of CircIGF2BP3 and IGF2BP3 at different stages of proliferation and differentiation and the ratio of the two transcripts at different stages of proliferation and differentiation. The Fug. 5A should show immunohistochemistry visualization of differentiation markers, MyHC. The authors must show the protein level of IGF2BP3 and CircIGF2BP3.
5) In Figure 6, the authors want to show that CircIGF2BP3 Promotes the Proliferation of Myoblast. To do this they transfect the chicken primary myoblasts with empty vector pCD2.1-ciR and pCD2.1-circIGF2BP3 (circlGFRBP3 overexpression vector) but do not show any data on the transfection efficiency. Transfection efficiency of Primary cells are generally relatively low. The data interpretation will be completely different if the transfection efficiency is 10% as compared to 90% transfection efficiency.
The authors do show the transcript level of cells transfected with pCD2.1-ciR and pCD2.1-circIGF2BP3 (Fig. 6D). The authors must also show the protein produced by CircIGF2BP3 if ANY to decipher the possible mechanism of how CircIGF2BP3 may regulate the myoblast proliferation and differentiation.
6) In Figure 7, the authors want to show that CircIGF2BP3 Promotes Myoblast Differentiation. They show that differentiation is increased by ~8%. Is this due to poor transfection efficiency or is this maximum possible enhancement of differentiation?
Tig. 7C: The expression of myogenic marker genes in myoblast overexpressing circIGF2BP3 is shown but there are lane markings? Is the first 3 lanes control cells and the other 3 from circIGF2BP3 overexpressing cells? Please label the lanes.
Need to show the peptide expressed from circIGF2BP3 if any so that we can interpretation the possible mechanism of how circIGF2BP3 regulates proliferation and differentiation of myoblast to myotubes; Sponge for miRNA, Transcriptional regulation, and Coding Potential.
Reviewer 2 Report
This study aims to characterize circIGF2BP3 structure and spatio-temporal distribution, and determine its role in skeletal muscle cells. The content is interesting for readers, but has several issues in the scientific writing and English as listed below:
line 40: Spell out SMSC and circTTN here.
line 41: competitive
line 42: Spell out circSEP3 here.
line 47: Spell out CDR1 and AGO.
line 48: Spell out IGF2BP3 here.
line 61: There are many grammatical errors in this manuscript, as shown in line 61-62 for instance. Authors should carefully check and correct English writing using professional editing service or something.
line 96-97: Methods for nuclear-cytoplasm separation does not appear anywhere. The procedure should be provided.
line 96-97: Methods for nuclear-cytoplasm separation does not appear anywhere. The procedure should be provided.
Figure 1: Letters in Figure 1 panels D-G are too small for readers to understand the illustrations. Enlarge the letters.
line 112: Please explain DNAMAN. If this is a public database, provide the information about it, such as URL.
Figure 2 legend is redundant, which should be brief and shortened.
Letters in the graphs are too small for readers to understand the illustrations. Enlarge the letters.
Figure 3 legend is redundant, which should be brief and shortened.
Letters in the graphs are too small for readers to understand the illustrations. Enlarge the letters.
In Figure 4, no error bar of SEM was indicated. Please add error bars.
This statement is quite unusual and is hardly understood. What do the authors mean by this? How was the proliferation measured? Please rewrite.
Panel A in Figure 5 observed by bright field image is very low quality. This should be replaced with phase-contrast images.
Letters in the graphs are too small for readers to understand the illustrations. Enlarge the letters.
Why was CCK-8 focused on? Please explain briefly.
Letters in the graphs are too small for readers to understand the illustrations. Enlarge the letters.
line 205-206: This sentence does not make sense. Rewrite.
As for Figure 7, I do not understand whether myogenic differentiation and myotube formation successfully progressed in the Panel A. Please provide the phase-contrast image. Letters in the graphs are too small for readers to understand the illustrations. Enlarge the letters.
line 242: Spell out PKP3 here.
line 247: veridate > validate
line 273: What sort of involvement do the authors hypothesize? Please discuss.
line 277: It is hard to understand the results of circIGF2BP3 expression in myoblast culture indicate that circIGF2BP3 seems to participate both proliferation and differentiation. Participation to both proliferation and differentiation sounds contradiction. How do the authors hypothesize the role of circIGF2BP3? Please discuss it.
line 280: Spell out circSVIL here.
line 281: Spell out circHIPK3 here.
line 284: There is no evidence that demonstrates the role of circIGF2BP3is different from those of circSVIL and circHIPK3.
There are many grammatical errors and careless mistakes in English writing. The manuscript should be thoroughly checked and corrected by professional native English writer, using commercial proofreading service.
Round 2
Reviewer 1 Report
The authors have already indicated in their manuscript (Line 58) “Current research has revealed that the downregulation of IGF2BP3 significantly delays the differentiation and induces proliferation of C2C12 myoblasts” which suggests that proliferation and differentiation are 2 pathways available for the myoblast and IGF2BP3 promotes differentiation and NOT proliferation. However, in the present manuscript the authors are suggesting that circIGF2BP3 promotes both proliferation and differentiation.
The authors have not addressed the following important questions.
I) The main point of the manuscript is that “circIGF2BP3 promotes the proliferation and differentiation” Based on literature IGF2BP3 promotes differentiation and maybe inhibit proliferation! If circIGF2BP3 promotes both proliferation and differentiation, the authors must explain how.
II) What is the role of circIGFBP3 in the expression of IGF2BP3 and IGF2? (Increase or decrease? and how is it done?)
Fig4: There is no correlation between expression of circIGF2BP3, IGF2BP3 and IGF2 with the expression of MyHC which increases gradually.
Expression of IGF2 and circIGF2BP3 is going up and down during the same period, sometime increasing together, eg. DM1, sometimes circIGF2BP3 is increased and IGF2 is decreased, eg. DM5.
Expression of IGF2BP3 is the same from 100%, DM1-4. This suggests that circIGF2BP3 does not regulate expression of IGF2BP3 or IGF2?
If circIGF2BP3 promotes differentiation it should be increasing during differentiation!
Fig. 5: A) The proliferation curve shows data up to 96hrs. D) Data for “48 h with circIGF2BP3 overexpression” E) Western Blot; How long after transfection? The authors must describe the experiment clearly. How many days after transfection were the cells tested for proliferation?
Fig. 6: I think the experiment is flawed. The author seems to have transfected the cells and prepared lysate “3 d of overexpression of circIGF2BP3”. Does it mean three days after transfection or three days of differentiation? The authors must describe the experiment clearly. How many days after transfection were the cells induced to undergo differentiation? How long were the cells in differentiation media before the lysate was prepared? The lysate for the western blot is from cells in Differentiation Media or proliferation media?
The authors show that transfection of pCD2.1- 226-circIGF2BP3 increase expression of MyHC (Fig. 6E) but don’t show the protein level of IGF2BP3 as shown in Fig. 5E! Without the expression data on IGF2BP3 (protein) we cannot determine the mechanism responsible for the increased expression of MyHC. The authors must show the expression level of IGF2BP3 in cells transfected with pCD2.1- 226-circIGF2BP3?
The relative myotube area is not the proper way to calculate differentiation. The accepted way to quantify differentiation is to calculate the Percentage of nuclei in myotubes.
Differentiation Index = (# of nuclei in myotubes/Total number of nuclei in image) X 100.
A nucleus is considered to be in a myotube if there are 3 or more nuclei in a MyHC stained myotube.
The authors have improved the manuscript, but it stills needs to be further improved before publication.
The manuscript is full of grammatical mistakes.
1) “Current research has revealed that the downregulation of IGF2BP3 significantly delays the differentiation and induces proliferation of C2C12 myoblasts, indicating that IGF2BP3 was closely related to animal muscle development and the myoblasts proliferation.” The sentence does not make any sense.
Can be rewritten as “indicating that IGF2BP3 plays a critical role in muscle development by promoting myoblasts proliferation”
2) There are still many grammatical errors and careless mistakes. eg. its "open reading frame" and NOT "opening reading frame"
3) Line 54: Insulin-like growth factor 2 mRNA binding protein 3 (IGF2BP3) is a member of the insulin-like growth factor mRNA binding protein (IGFBP) family, which can bind to insulin-like growth factor 2 (IGF2). To the IGF2BP3 mRNA or protein.
The manuscript should be checked and corrected by a native English speaker/writer as there are too many mistakes in language. I suggest that the authors use some professional services to improve the manuscript.
Reviewer 2 Report
line 195: "50% proliferation and 100% proliferation" no one understand this statement. How do authors define proliferation quantitatively? Does this mean 50% and 100% confluence? Please explain.
In the manuscript, I do not find any explanation about what Figure 4A. Panel A in Figure 5 observed by bright field image is very low quality. This should be replaced with phase-contrast images. What do the authors want to show by this figures? There is no reader who can understand the images and what the figures mean. If the authors do not want to show anything by the panel A and cannot replace the panel with phase-contrast images, this panel should be omitted, although images of myotube formation during myogenic differentiation is normally necessary.In addition, use of magnification is inappropriate to indicate the size of cells.Scale bars should be put in each of images in the presented figures.
There are several grammatical errors and careless mistakes in English writing. Please check the text carefully.
Round 3
Reviewer 1 Report
Need to improve the English in the manuscript
Line 55: It regulats..." Check spelling.
Line 85: analysis showed that circIGF2BP3 was originated from" Check sentence
Line 88: "cicrIGF2BP3", Check spelling
Line 163: "significantly increased", should be "significantly higher"
Author Response
Dear Editors and Reviewers:
We highly appreciate your valuable comments concerning our article entitled “Circular RNA circIGF2BP3 promotes the proliferation and differentiation of chicken primary myoblasts” (Manuscript ID: ijms-2611812). Those suggestions are all constructive and helpful for revising and improving our paper. We have amended the manuscript carefully according to reviewers’ advice and comments. The modified descriptions were highlighted with Tracked Changes in our manuscript. The “point-by-point” response to each comment are as follows.
Best wishes,
Shudai Lin
Reviewer 1#
Comments and Suggestions for Authors
Need to improve the English in the manuscript
Response: Thank you for your comments. We have strived to amend the whole manuscript.
Comments on the Quality of English Language
Line 55: It regulats..." Check spelling.
Response: Your suggestion is highly appreciated. We have corrected this word.
Line 85: analysis showed that circIGF2BP3 was originated from" Check sentence
Response: We really appreciate for your suggestion. We have modified this sentence based on your comments.
Line 88: "cicrIGF2BP3", Check spelling
Response: Thank you for your valuable suggestion. We have rewritten this word as you suggested.
Line 163: "significantly increased", should be "significantly higher"
Response: Thank you for your comments. We have modified this point based on your suggestions.
Reviewer 2 Report
The present manuscript has improved and satisfies the minimal level for publication.
The manuscript had many grammatical errors at the submission. Even if most of them were corrected, the authors need to carefully check overall thoroughly and correct if necessary.
